# Detection of Beta-Lactam-Resistant *Escherichia coli* and Toxigenic *Clostridioides difficile* Strains in Wild Boars Foraging in an Anthropization Gradient

**DOI:** 10.3390/ani11061585

**Published:** 2021-05-28

**Authors:** Laila Darwich, Chiara Seminati, Jorge R. López-Olvera, Anna Vidal, Laia Aguirre, Marina Cerdá, Biel Garcias, Marta Valldeperes, Raquel Castillo-Contreras, Lourdes Migura-Garcia, Carles Conejero, Gregorio Mentaberre

**Affiliations:** 1Departament de Sanitat i Anatomia Animal, Universitat Autònoma de Barcelona (UAB), CP-08193 Cerdanyola del Vallès, Spain; annavordeig@gmail.com (A.V.); laia.aguirre@uab.cat (L.A.); marina.cerda.aunon@gmail.com (M.C.); biel.garcias@uab.cat (B.G.); 2IRTA, Centre de Recerca en Sanitat Animal (CReSA, IRTA-UAB), Campus de la Universitat Autònoma de Barcelona, CP-08193 Cerdanyola del Vallès, Spain; Lourdes.migura@irta.cat; 3OIE Collaborating Centre for the Research and Control of Emerging and Re-emerging Swine Diseases in Eu-rope (IRTA-CReSA), Bellaterra, CP-08193 Barcelona, Spain; 4Wildlife Ecology & Health Group (WE&H) and Servei d’Ecopatologia de Fauna Salvatge (SEFaS), Departament de Medicina i Cirurgia Animals, Universitat Autònoma de Barcelona (UAB), CP-08193 Cerdanyola del Vallès, Spain; jordi.lopez.olvera@uab.cat (J.R.L.-O.); marta.valldeperes@uab.cat (M.V.); castilloraquel91@gmail.com (R.C.-C.); carles.conejero@uab.cat (C.C.); gregorio.mentaberre@udl.cat (G.M.); 5Departament de Ciència Animal, Escola Tècnica Superior d’Enginyeria Agraria (ETSEA), Universitat de Lleida (UdL), CP-25098 Lleida, Spain

**Keywords:** antimicrobial resistance, ESBL, AmpC, *Clostridiodes difficile*, *Escherichia coli*, *Sus scrofa*, wild boar, urban areas, zoonotic agents

## Abstract

**Simple Summary:**

The wild boar (*Sus scrofa*) has been identified as a reservoir of zoonoses and food-borne pathogens. Wild boar populations are growing worldwide, also near urban centers such as Barcelona (Catalonia, Spain). The aim of this study was to assess the presence of zoonotic agents in the wild boar populations from the Metropolitan Area of Barcelona (MAB). The detection of *Escherichia coli* carrying critical antimicrobial resistance genes to β-lactams and, for the first time in Spain, the detection of toxigenic *Clostridioides difficile* strains in wild boars foraging in urban areas show the value of this game species as a sentinel of antimicrobial-resistant bacteria (AMRB) and zoonotic agents in the environment. Moreover, the wild boars foraging in urban and peri-urban locations were more exposed to AMRB sources than the wild boars dwelling in natural environments. The responsible health agencies should undertake specific actions to fully assess the potential implications for the human population and/or environmental health. Zoonotic diseases and hazards beyond the compulsory surveillance diseases should be specifically targeted in the growing human–wildlife interface of urban environments such as the MAB under the One Health approach.

**Abstract:**

Disease transmission among wild boars, domestic animals and humans is a public health concern, especially in areas with high wild boar densities. In this study, fecal samples of wild boars (*n* = 200) from different locations of the Metropolitan Area of Barcelona were analyzed by PCR to explore the frequency of β-lactamases and extended cephalosporin and carbapenem resistance genes (ESBLs) in *Escherichia coli* strains and the presence of toxigenic *Clostridioides difficile*. The prevalence of genes conferring resistance to β-lactam antimicrobials was 8.0% (16/200): *bla*_CMY-2_ (3.0%), *bla*_TEM-1b_ (2.5%), *bl*a_CTX-M-14_ (1.0%), *bla*_SHV-28_ (1.0%), *bla*_CTX-M-15_ (0.5%) and *bla*_CMY-1_ (0.5%). *Clostridioides difficile* TcdA+ was detected in two wild boars (1.0%), which is the first report of this pathogen in wild boars in Spain. Moreover, the wild boars foraging in urban and peri-urban locations were more exposed to AMRB sources than the wild boars dwelling in natural environments. In conclusion, the detection of *E. coli* carrying ESBL/AmpC genes and toxigenic *C. difficile* in wild boars foraging in urban areas reinforces the value of this game species as a sentinel of environmental AMRB sources. In addition, these wild boars can be a public and environmental health concern by disseminating AMRB and other zoonotic agents. Although this study provides the first hints of the potential anthropogenic sources of AMR, further efforts should be conducted to identify and control them.

## 1. Introduction

Wildlife can contribute to preserving and disseminating pathogens and antimicrobial resistance (AMR) genes shared with livestock and humans [1,2]. In recent years, the study in wildlife of genes conferring resistance to human last-resort antibiotics, such as extended-spectrum β-lactamases (ESBLs), AmpC-type β-lactamases (AmpC) and carbapenem resistance genes, has gained relevance [3]. In spite of the strong hunting pressure, wild boar (*Sus scrofa*) populations are rising steadily worldwide [4,5], which is an increasing cause of concern since the wild boar has been identified as a reservoir of zoonoses and food-borne pathogens [6,7,8,9,10,11]. On the other hand, the wild boar is a suitable sentinel to search shared pathogens and environmental AMR contamination, due to its opportunistic feeding habits and rooting behavior [12], in addition to its abundance and wide distribution. A behavioral adjustment of wildlife to the urban environment is the use of anthropogenic food resources [13], as described in wild boars from the Metropolitan Area of Barcelona (MAB) [14,15]. Furthermore, socioeconomic variables and human attitudes are key in the availability of anthropogenic food resources for wild boars in urban and peri-urban areas [15,16,17].

The MAB is located in north-eastern Spain and comprises 36 municipalities with an average density of 5093 inhabitants per 100 ha [18]. However, neither the conditions nor the human population distribution are homogeneous within the MAB (Figure 1). First, Collserola Natural Park (CNP) is a roughly 8000 Ha Mediterranean mountain massif located within the MAB that includes uninhabited areas where wild boar hunting is allowed (Figure 1, CN and CS) and residential areas [19]. Second, the western districts of Barcelona (16,150 inhabitants per 100 ha) [20], adjacent to CNP, are frequently visited by wild boars [14] and can be further classified into two areas: the north-western area (Figure 1, BN; ~17,100 inhabitants/100 ha), which holds human health complexes and residences for the elderly, with socioeconomic characteristics leading to higher availability of anthropogenic food sources for wild boars [17]; and the south-western area (Figure 1, BS), with a lower human population density (~8900 inhabitants/100 ha) and a more negative attitude of residents towards wild boars [17]. Last, the campus of the Universitat Autònoma de Barcelona (UAB) is located north of CNP with two separate areas differentiated: a north-western part (Figure 1, UN) with a livestock farm and the Veterinary Teaching Hospital of the Veterinary School, and a south-eastern one (Figure 1, US), with no potential anthropogenic sources of AMR identified.

In Spain, infection with *C. difficile* is the most common nosocomial infection and cause of antibiotic-associated diarrhea in hospitalized patients, with a high economic impact for the Spanish National Health System [21]. On the other hand, *C. difficile* is one of the most important uncontrolled causes of neonatal diarrhea in pig production, affecting one-week-old piglets in Catalonia [22], with a significant impact on the health and productivity of domestic pigs. However, there is scarce information about *C. difficile* infection in their wild ancestor. Therefore, epidemiological data are needed to assess the role of wild boars as potential reservoirs of this zoonotic pathogen in Spain.

To increase the knowledge on AMR bacteria (AMRB) presence and dissemination pathways in the MAB, we assessed the occurrence of ESBLs, AmpC, carbapenem-resistant enterobacteria and toxigenic *C. difficile* strains in fecal samples of wild boars from three locations in the MAB with different degrees of urbanization, human activity and, consequently, potential anthropogenic sources of AMR. 

## 2. Materials and Methods

### 2.1. Animal Sampling

From 2015 to 2019, fecal samples from 200 wild boars (102 females and 98 males, ranging from three months to nine years of age, determined through tooth eruption and wear) were collected and analyzed at the Infectious Diseases Unit of the Veterinary School of the UAB. The wild boars in CNP were hunted in natural areas (*n* = 103, 35 hunting events). The wild boars in urban and peri-urban areas were captured and euthanized through tele-anesthesia with a blowpipe by a veterinarian in Barcelona (contracts 15/0174, 16/0243 and 16/0243-00-PR/01 with Ajuntament de Barcelona—Barcelona City Council, *n* = 63, 37 capture events) or using cage traps prior to anesthesia in the UAB (authorization AC/190 from Generalitat de Catalunya—Government of Catalonia, *n* = 34, 17 capture events), as previously reported [23]. All the wild boar capture and hunting operations were carried out as part of the regular population management plan for the species, not for research, and according to national and local legislation. The feces were collected directly from the rectum, stored in a sterile container, placed in refrigeration and sent to the laboratory within 24 h.

### 2.2. Microbiological and Molecular Identification of Pathogens

For *E. coli* isolation, the fecal samples were directly cultured on Columbia blood agar (BD GmBh, Munich, Germany) and MacConkey agar (Oxoid, UK), not supplemented and supplemented with ceftriaxone (1 mg/L), and aerobically incubated during 24 h at 37 °C. All the strains were confirmed as *E. coli* using conventional biochemical tests or the API system (bioMérieux, Marcy l’Etoile, France). For *C. difficile* isolation, the samples were firstly treated with ethanol (96%) for 35 min to eliminate the vegetative cells and then centrifuged (8000× *g*) as previously described [24]. The pellet was then cultured on a selective medium *C. difficile* agar base (Condalab, Madrid, Spain) and incubated anaerobically for 48 h at 37 °C.

For molecular identification, DNA was extracted by boiling the bacterial growth from the MacConkey cultures supplemented with ceftriaxone (1 mg/L) and from *Clostridium* spp. selective medium plates as previously described [25,26]. Briefly, all the bacterial growths were diluted in 600 μL of sterile distilled water, and 200 μL of the dilution was then transferred to a new tube. Two-hundred microliters of sterile distilled water was added to each tube. The tubes were boiled in a water bath for 10 min and then centrifuged at 13,000 rpm for 5 min. After centrifugation, the supernatant was recovered and stored at −80 °C until processing.

The detection of toxigenic *C. difficile* strains was performed with a standardized PCR protocol for TcdA and TcdB toxins as previously described [22]. The PCR program consisted of 10 min at 94 °C, followed by 25 cycles of 50 s at 94 °C, 40 s of annealing at 53 °C and 50 s of extension at 72 °C, with a final extension step of 3 min at 72 °C.

### 2.3. Antimicrobial Resistance Genes

β-lactamase—*bla*_TEM,_ *bl*a_CTX,_ *bla*_SHV_, *bla*_CMY-1,_
*bla*_CMY-2,_
*bla*_OXA_, *bla*_VIM_ and *bla*_CLR_—genes were investigated using previously described PCR protocols [27]. Briefly, PCR conditions were homogenized for all the reactions as follows: 5 min at 94 °C, followed by 25 cycles of 1 min at 94 °C, 1 min of annealing at 55 °C and 1 min of extension at 72 °C, with a final extension step of 7 min at 72 °C. Controls for all genes were provided by Dr. Migura (IRTA-CReSA, Bellaterra, Spain) and by the Veterinary Diagnostic Laboratory in Infectious Diseases of the Universitat Autònoma de Barcelona (UAB, Bellaterra, Spain).

Sanger DNA sequencing was conducted for *bla*_TEM,_ *bl*a_CTX_ and *bla*_SHV_ PCR products at the Genomic and Bioinformatics Service of the Universitat Autònoma de Barcelona (Spain). Sequences and chromatograms were manually explored to trim bad-quality bases with BioEdit 7.2. Once the assembly of the consensus sequences was conducted, both complete and partial sequences were aligned using the Clustal Omega program and finally blasted against the public database (National Center for Biotechnology Information, NCBI, Bethesda, MD, USA).

### 2.4. Statistical Analysis

All the statistical analyses were performed with R software, version 3.6.0 (R Development Core Team, Auckland, New Zealand, 2019). The presence of the studied AMR genes was compared among sampling areas by means of χ2 tests (catdes package). Generalized linear mixed-effects models (lme4 package to fit the models, and MuMIn package to select the best model) were used to correct for potential non-independency of observations in the wild boars captured in the same capture events. 

## 3. Results

Sixteen out of two hundred wild boars analyzed (8.0%) were positive for β-lactam-resistant *E. coli* strains, with *bla***_CMY-2_** (3%), *bla*_TEM-1b_ (2.5%) and *bla*_CTX-M_ (1.5%) being the most commonly detected genes (Table 1). The sequencing analysis confirmed the presence of ESBL-producing *E. coli* strains in five wild boars, expressing *bla*_CTX-M-14_ (*n* = 2), *bla*_CTX-M-15_ (*n* = 1) and *bla*_SHV-28_ (*n* = 2) variants. 

β-lactam resistance genes were more frequently isolated from the urban/peri-urban wild boars—Barcelona and UAB—than in the forest ones: CNP (*p* < 0.05; Table 2). 

At a finer spatial scale, AMR detection was significantly higher in both north-western areas defined within Barcelona (0.05 < *p*-value < 0.1) and the UAB (*p* < 0.05), with a higher potential exposure to anthropogenic sources of AMR, identifying two spatial clusters of positivity to β-lactam-resistant E. coli strains (Figure 1). 

No resistance to carbapenems was detected associated with the *bla*_OXA_, *bla*_VIM_ and *bla*_CLR_ genes. Regarding the presence of toxigenic *C. difficile* strains, two wild boars (1.0%) were positive for toxin A (TcdA) of *C. difficile* by PCR.

## 4. Discussion

This study reports extended-spectrum β-lactamase (ESBL)-producing *E. coli* (*bla*_CTX-M-14_, *bla*_CTX-M-15_ and *bla*_SHV-28_) strains and the zoonotic bacteria *C. difficile* for the first time in wild boars in Spain. The emergence of ESBL-producing *E. coli* in human patients is largely due to the spread of *bla*_CTX-M_, especially the *bla*_CTX-M-15_ variant. In Spain, *bla*_CTX-M-14_ and *bla*_CTX-M-15_ are the most predominant ESBL types found in hospitalized human patients [28]. Moreover, in recent years, pigs and pork meat have also been described as a reservoir of human pathogenic ESBL-producing *E. coli* and a reservoir of transferable *bla*_CTX-M-14_ and *bla*_CTX-M-15_ genes, which could potentially be transferred to close-contact humans [29]. On the other hand, *bla*_SHV-28_ has been reported in enterobacteriaceae with reduced susceptibility to third-generation cephalosporins recovered from diseased dogs and cats in Europe [30] and from asymptomatic wild mammals and birds in Catalonia [27]. The detection of *bla*_TEM-1b_ as the most frequent gene conferring resistance to β-lactam antimicrobials agrees with previous reports in wild boars in Portugal and other parts of Europe [31,32]. However, to the best of our knowledge, carbapenem resistance has not been previously reported in wild boars in Spain, and none of the examined wild boars from the MAB were positive. Although OXA-48 has been reported in wild boars in Algeria [33] and in wild hedgehogs and mustelids in Catalonia [27], it has not been identified in wild boars from Catalonia. 

*Clostridioides difficile* is a well-established pathogen of both humans and animals that contaminates foods and the environment [34]. In humans, *C. difficile* is the principal cause of antibiotic-associated diarrhea, emerging as a major infection associated with health care [21]. Genomic analyses have shown an overlap of *C. difficile* isolates from animals and people, suggesting that a zoonotic reservoir may contribute to recurrence [34]. Livestock and specifically domestic pigs are a major reservoir and amplification host for *C. difficile*, including lineages of clinical importance. Moreover, this infection can clearly cause severe disease in young pigs; in particular, it has been associated with pseudomembranous colitis and diarrhea outbreaks in neonatal pigs [22]. Regarding wild pigs as reservoirs, *C. difficile* has been previously reported in feral pigs of North Carolina (4%) [35]. To the best of our knowledge, this is the first report of toxigenic *C. difficile* in wild boars in Spain and Europe. The lower prevalence observed here as compared to North Carolina could be related to the age of the wild boars sampled, mostly (171 out of 200) over six months, since in domestic pigs, *C. difficile* is mostly detected in one-week-old piglets [36], significantly declining with age [37].

The spatial pattern observed suggests possible acquisition pathways for the wild boar and reinforces the starting hypothesis of urban environments and anthropogenic inputs as sources of AMR bacteria. The wild boars captured in Barcelona and the UAB campus are more exposed to anthropogenic food sources, including waste, while the wild boars hunted in CNP maintain a wilder nature and trophic behavior [15]. In addition to the presence of important health complexes (e.g., Hospital Vall d’Hebron) and a higher density of the population, the north-western districts of Barcelona (BN) have specific socioeconomic conditions (i.e., higher density of inhabitants, lower income per capita). This results in a higher access of wild boars to anthropogenic food sources, including easier access to rubbish, direct feeding and generally more positive attitudes by inhabitants towards the presence of this species in their neighborhoods [16,17]. Regarding the UAB, the wild boars captured closer to livestock and veterinary facilities had the highest frequency of AMR genes. Both urbanization (and in this case, a higher population density) and suspected anthropogenic sources of AMR have been previously identified as risk factors for exposure to other viral and bacterial pathogens, as well as to environmental pollutants, in the wild boar population from the MAB [11,38,39]. In Catalonia, a higher prevalence of antibiotic-resistant bacteria (*E.coli* and *Enterococcus* spp.) has been reported in urban wild boars than in their rural counterparts, suggesting that urban wild boars may be more exposed to certain antibiotic-resistant bacteria or antibiotic resistance genes [40]. Our findings have implications for AMR dissemination in the wild boar population, other sympatric wildlife populations, the environment and potential dissemination pathways back to humans as a cause of public health concern that should not be neglected. This study not only showed that urban and peri-urban wild boars are more exposed to AMR but also identified potential factors driving a higher exposure risk for urban wildlife, such as health premises and social factors such as the human population density and lower income. The crepuscular and nocturnal foraging and exploratory behavior of wild boars allows them to contaminate with their feces places that people and pets use in the daytime [41]. 

On the one hand, the results of this study contribute to evidence that *bla*_CMY-2_, *bl*a_CTX-M-14_ and *bla*_CTX-M-15,_ found in 8.0% of the wild boars from the MAB, are the most widespread ESBLs around the world, being reported in all ecological niches: humans, animals and the environment [42,43]. On the other hand, the detection of toxigenic *C. difficile* strains for the first time in wild boars foraging in urban areas shows the value of this game species as a sentinel for AMRB and zoonotic agents in the environment. Hence, the presence of AMRB in the wild boar population from the MAB may pose a risk of increased environmental dissemination due to their amplification potential. The direct and indirect exposures of the wild boars from CNP to anthropogenic AMR sources may pose an additional risk for hunters and hunting dogs through handling of wild boar carcasses or consumption of inadequately cooked meat. In urban areas, the presence of zoonotic bacterial species in wildlife is especially concerning because the human population potentially at risk is considerably bigger, beyond the usual social groups in contact with wildlife. This is especially noticeable for the MAB and, specifically, for Barcelona. Since the urban wild boar phenomenon is currently emerging in a number of cities all around the world, our results may deserve interest beyond the local context of our study. 

## 5. Conclusions

In conclusion, the presence of *E. coli* carrying ESBL and AmpC genes and toxigenic *C. difficile* in wild boars from the MAB shows the potential value of this game species as a sentinel of AMR and zoonotic agents in the environment due to its regular management and easy access to samples. The responsible institutions should address specific actions under the One Health approach to fully assess the potential implications of these findings for the human population and/or for environmental health. Zoonotic diseases and hazards beyond the compulsory surveillance diseases in wildlife should be specifically targeted in urban and peri-urban scenarios such as the MAB, where a new wildlife–human interface is growing. Since wild boars may become infected and/or acquire resistant bacteria through the consumption of uncollected waste in the public space of cities, biosecurity preventive strategies for reducing the contact between these wild species and humans should be further developed.

## Figures and Tables

**Figure 1 animals-11-01585-f001:**
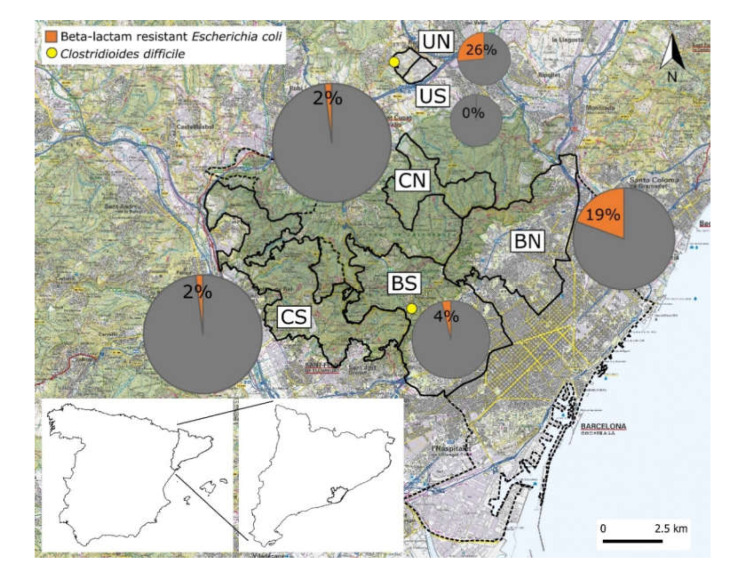
Study areas including the north-western (BN) and south-western (BS) districts of Barcelona bordering Collserola Natural Park (CNP; the green shaded area), north-eastern (CN) and south-western (CS) areas of CNP where wild boar hunting is allowed and the north-western (UN) and south-eastern (US) parts of the campus of the Universitat Autònoma de Barcelona. The dashed line shows the limit of the municipality of Barcelona. All the study areas are encompassed in the Metropolitan Area of Barcelona, whose position within the Catalonia region in the north-east of Spain is shown in the lower left corner of the figure. Pie charts show the occurrence of wild boars positive for AMR genes with the size being proportional to the number of wild boars sampled in the corresponding study areas.

**Table 1 animals-11-01585-t001:** Number and frequencies of wild boars positive for β-lactamase genes.

AMR Genes	Number of Positives	Frequency	Confidence Interval
	(*N* = 200)	(%)	95%
*bla* _CMY-1_	1	0.5	0.1–2.8
*bla* _CMY-2_	6	3.0	1.4–6.4
*bla* _CTX-M-14/M-15_	3	1.5	0.5–4.3
*bla* _SHV-28_	2	1.0	0.3–3.6
*bla* _TEM-1b_	5	2.5	1.1–5.7
Total	16 *	8.0	5.0–12.6

* A single sample was positive for two genes (*bla*_CTX-M-14_ and *bla*_CMY-2_).

**Table 2 animals-11-01585-t002:** Number and frequencies of wild boars positive for genes conferring resistance to β-lactam antimicrobials according to origin.

Origin	Positive/Sampled	Frequency (%)	Confidence Interval 95%
Urban/peri-urban wild boars	14/97	14.4 ^α^	7.4–21.4
Barcelona *	9/63	14.3 ^A^	5.6–22.9
BN	7/36	19.4 ^a^	6.5–32.4
BS	1/26	3.8 ^b^	0.0–11.2
University campus (UAB)	5/34	14.7 ^A^	2.8–26.6
UN	5/19	26.3 ^a^	6.5–46.1
US	0/15	0.0 ^b^	-
Forest wild boars (CNP)	2/103	1.9 ^β,B^	0.0–4.6
Total	16/200	8.0	4.2–11.8

BN: north-western districts of Barcelona; BS: south-western districts of Barcelona; UAB: Universitat Autònoma de Barcelona; UN: north-western part of the UAB campus; US: south-eastern part of the UAB campus; CNP: Collserola Natural Park. Different superscripts within the same level (α and β; A and B; a and b) indicate significant differences between groups (*p* < 0.05, except for BN and BS [0.05 < *p*-value < 0.1]). * One of the wild boars from Barcelona was sampled out of the two areas defined (Figure 1).

## Data Availability

Data will be made available upon reasonable request to the corresponding author.

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
