# Peer review of "Detection of Beta-Lactam-Resistant Escherichia coli and Toxigenic Clostridioides difficile Strains in Wild Boars Foraging in an Anthropization Gradient"

_animals, 2021, doi:10.3390/ani11061585_

Round 1

Reviewer 1 Report

This revised version is much improved.

Author Response

Thank you very much for your positive report.

Reviewer 2 Report

The changes made during revision have significantly improved the manuscript and cleared up most of my concerns. There are still some points that should be addressed, after which the mansucript should be acceptable for publiaction.

The sequencing revealed the presence of the blaTEM-1b gene in almost a third of the isolates. You have already rewritten parts of the manuscript accordingly, however, the presentation of the results needs to be adjusted in some additional parts of the manuscript, which are currently still referring specifically to cephalosporin-resistance/ESBL despite including blaTEM-1b (e.g. l. 175, l. 187.). Please check your whole results and discussion sections accordingly.

In addition, there are still too many grammatical errors/typos in the manuscript and the language quality is not very good. Further proofreading is necessary and a linguistic revision by a native speaker is strongly recommended. A few examples include: l. 29 "antimicrobial resistance bacteria" - should be "resistant"; l. 40 should be "resistance genes"; l. 95 "on health" etc.). Some more examples are mentioned below.

L. 40 - 43: These two sentences sound a bit misleading. Please either include non-extended spectrum beta-lactamases as a target for this study in the first sentence (in addition to cephalosprin and carbapenem resistance) or remove TEM-1b from the list in the second sentence. Otherwise it appears to implicate that it confers one of the resistances mentioned in the previous sentence.

L. 49: general statement: "good sentinel of environmental health" - please be specific about its value as a sentinel for antimicrobial resistance in wild animals. Even if there was no resistance found, it would not necessarily indicate an overall healthy environment.

L. 145-147: Please rewrite to indicate that blaTEM, blaSHV and blaOXA are not necessarily ESBLs/carbapenemases. I would suggest to simply refer to them as beta-lactamase genes at this point and later state that the PCR products were sequenced in order to identify ESBL/carpanemase variants of these genes.

L. 167: please use beta-symbol as you have previously in the introduction

L. 168: Move "being" after "blaCTX-M (1,5%)"

L. 172-173: Please change title to include non-ESBL beta-lactamases as you list TEM-1b in the table (similar to title of Table 2). Also please use beta-symbol in both Table headings

L. 186: "exposure"

L. 202: Please use beta-symbol

L 211: the authors of the cited study reported on isolates with "non-wild type susceptibilty", i.e. reduced susceptibility

L. 213: Please include a sentence referring to the specific TEM-variant in your study.

L. 228: missing "s" in "wild pigs"

L. 254 ff.: Please focus the discussion more on the relevance of your findings (i.e. the presence of beta-lactamse resistant E. coli and the identification of genes, specific risk factors; C. difficile). A previous study was already published about general AMR in wild boars in the MAB (Navarro-Gonzalez et. al 2018, doi: 10.1007/s10344-018-1221-y). This study is not mentioned so far at all, despite its obvious relevance to the current study. It should be made clearer what knowledge gaps were present, how this study builds upon existing knowledge and what novel insights the current study provides. This also pertains to other parts of the manuscript such as l. 99 ff.

Author Response

This manuscript is a resubmission of an earlier submission. The following is a list of the peer review reports and author responses from that submission.

Round 1

Reviewer 1 Report

This study examined 200 wild boar fecal samples collected from MAB area, Spain, and reported the prevalence of antimicrobial resistance bacteria (AMRB) discovered from these fecal samples. Although the information is important, following concerns need to be addressed before this article can be considered for publication:

  1. Collection and storage methods of fecal sample before analysis, and the age-group and gender ratios of these 200 wild boars need to be provided in “2.1 Animal samples”.
  2. The prevalence for all targets were low, between area (environments) comparison using prevalence (%) may not be appropriate, please provide justification.
  3. The conclusion of wild boar can be a good sentinel of AMRB in the environment and the proposition of conducting compulsory surveillance in urban area requires more practical suggestions, especially with the low prevalence found by the present study.

Reviewer 2 Report

This study examined the presence of E. coli carrying beta-lactam resistance genes as well as of TcdA-positive C. difficile in feacal samples of wild boars in the Barcelona metropolitan area. The detection of C. difficile is interesting. The data regarding E. coli in its current state is lacking and the conclusions are not necessarily fully supported by the results. Most importantly, no distinction is made between ESBLs and non-extended spectrum beta-lactamases. Specifically blaTEM and blaSHV are not necessarily ESBLs but quite often "regular" BLs. Sequence analyses would have been necessary to identify them as ESBL-variants of the respective genes. There is also no data confirming a resistance phenotype of the isolates, hence there is a possibility that some of the isolates may in fact not even be cephalosprin-resistant. It is either necessary to confirm their ESBL-status by sequence analyses, ideally coupled with susceptibility testing, or to substantially revise the text to reflect these uncertanties. The overall writing needs to be improved and a linguistic revision is necessary.

Abstract: Very abrupt beginning. Please give a short introductory sentence.

Introduction: The first paragraph nicely introduces the topic of AMR in wild boars and their possible roles as a reservoir and sentinels. The second, shorter paragraph (L. 71-75), describing the aim of the study, should be moved to the end of the introduction. The authors might also want to consider moving preceding lines 69-70 there as well. The third paragraph, discussing the Metropolitan Area of Barcelona is very detailed and thus rather hard to follow. More importantly, it does not include a single reference despite making up almost half of the introduction. At the very least, there should be a reference for the population data. I suggest significantly shortening this paragraph as it partly repeated in the discussion and focus more on C. difficile. Why was C. difficile chosen as a target for this study?

In the materials and methods section, important aspects are missing, which would be crucial to fully evaluate the results. Most importantly, it is necessary to describe the isolation process in more detail. There is no mention of a selective enrichment step or selective plating on media containing cephalosporins. It remains unclear what was used for DNA extraction (single colonies?). If only single colonies were picked from unselectively grown E. coli cultures, it cannot be ensured that all ESBL/AmpC-producers present in the samples were detected if there were potentially also non-ESBL/AmpC-producers in the same sample.

Results & Discussion: It needs to be clearly established what data was collected and what conclusions can be drawn from it. Avoid generalizations and statements that are not supported by your data. The discussion is very short and C. difficile is hardly mentioned (the meaning of its presence, potential risks etc. should be discussed). As stated regarding the introduction, information about the socioeconomic and demographic conditions in the study area are not supported by references. While I realize that finding references about this may be difficult, it features so prominently in the conclusions drawn from the data that there is a need to provide at least some references or objective data supporting these statements.

Specific comments:

L. 62: ...as "a" reservoir for zoonoses...

L. 69-70: Please elaborate on this. What relevant observations have been made regarding the influence of these factors on the availability food?

L. 81: Please don't start sentence with an abbreviation.

L. 95: If only faecal samples were collected, and no live animals were involved in the study, please change the heading of this paragraph (i.e. "Sampling" or "Feacal samples").

L. 96-99: Please specify who conducted the capture and euthanasia events. Also, the actual sampling needs to be described in more detail (when and how were samples taken, how were they stored, how long until analyses etc.),

L. 101:  Again, important details are missing about the microbiological analyses (were feacal samples directly plated? Were they diluted? How much material was used? etc.). Specify "conventional biochemical tests" and provide reference.

L. 120: Resistance genes, not resistant genes. Revise sentence, beginning and ending do not fit together.

L. 120-125: I am wondering how all these different primers can be used at the same annealing temperature if not specifically designed for this purpose. The references do not themselves describe the primers or their original annealing temperatures and PCR protocols. Please include original references. Were controls available for all genes to ensure the PCR protocol used was suitable for all primers (if so, were they reference strains or field isolates confirmed as positive)?

L. 121: Use italics for "bla" and subscript type designation when reffering to genes (e.g. blaCTX-M). Check throughout manuscript.

L. 128: "AMR altogether" - general antimicrobial resistance was not evaluated in this study. Generally, only the presence of resistance genes was determined, no phenotypic data was generated. Please be specific about your gathered data and avoid inappropriate generalizations.

L.136/Table 1: See above. TEM and SHV are not confirmed as ESBLs.

L. 166: Which previously unreported AMR genes are you referring to?